# Z-Schemed WO$_3$/rGO/SnIn$_4$S$_8$ Sandwich Nanohybrids for Efficient Visible Light Photocatalytic Water Purification

**Pingfan Xu** [1,2], **Siyi Huang** [2], **Minghua Liu** [2,*], **Yuancai Lv** [2], **Zhonghui Wang** [1], **Jinlin Long** [3], **Wei Zhang** [4] **and Haojun Fan** [1,*]

1. National Engineering Laboratory for Clean Technology of Leather Manufacture, Sichuan University, Chengdu 610065, China; shoppingfan@126.com (P.X.); wangzhonghui652@163.com (Z.W.)
2. College of Environment & Resource, Fuzhou University, Fuzhou 350116, China; hsy920329@163.com (S.H.); yclv@fzu.edu.cn (Y.L.)
3. State Key Lab of Photocatalysis on Energy and Environment, College of Chemistry, Fuzhou University, Fuzhou 350116, China; jllong@fzu.edu.cn
4. State Key Laboratory of Polymer Materials Engineering, Polymer Research Institute at Sichuan University, Chengdu 610065, China; weizhang@scu.edu.cn
* Correspondence: mhliu2000@fzu.edu.cn (M.L.); fanhaojun@scu.edu.cn (H.F.)

**Abstract:** Semiconductor photocatalysis has received much attention as a promising technique to solve energy crisis and environmental pollution. This work demonstrated the rational design of "sandwich" WO$_3$/rGO/SnIn$_4$S$_8$ (WGS) Z-scheme photocatalysts for efficient purification of wastewater emitted from tannery and dyeing industries. Such materials were prepared by a combined protocol of the in situ precipitation method with hydrothermal synthesis, and structurally characterized by XRD, SEM, HRTEM, UV-vis DRS, and PL spectroscopy. Results showed that the Z-schemed nanohybrids significantly enhanced the photocatalytic activity compared to the single component photocatalysts. An optimized case of the WGS-2.5% photocatalysts exhibited the highest Cr(VI) reduction rate, which was ca. 1.8 and 12 times more than those of pure SnIn$_4$S$_8$ (SIS) and WO$_3$, respectively. Moreover, the molecular mechanism of the enhanced photocatalysis was clearly revealed by the radical-trapping control experiments and electron paramagnetic resonance (ESR) spectroscopy. The amount of superoxide and hydroxyl radicals as the major reactive oxygen species performing the redox catalysis was enhanced significantly on the Z-scheme WGS photocatalysts, where the spatial separation of photoinduced electron–hole pairs was therefore accelerated for the reduction of Cr(VI) and degradation of Rhodamine B (RhB). This study provides a novel strategy for the synthesis of all-solid-state Z-scheme photocatalysts for environmental remediation.

**Keywords:** WO$_3$/rGO/SnIn$_4$S$_8$; Z-scheme photocatalyst; active oxygen species; photocatalytic Cr(VI) reduction; rGO

## 1. Introduction

Heterogeneous photocatalysis is a potential technique which provides an environmentally friendly way to solve energy crises and environmental pollution [1,2]. Over the past few decades, it has been widely applied to water splitting, CO$_2$ reduction, organic photosynthesis, and pollutant degradation; however, its efficiency is limited by the poor light absorption and high charge recombination of semiconducting photocatalysts [3–5]. Many strategies, including dye sensitization or grafting [6], modifying the band gap edge by co-doping [7], and facilitating charge separation by integrating plasmonic metals [8,9], have already been proposed to create visible (even near infrared) light

response photocatalysts to overcome these intrinsic disadvantages [10]. Moreover, constructing type II nanoheterojunctions or molecular junctions at the interface of semiconducting photocatalysts was well established as another promising approach to enhance the efficiency of various photocatalytic reactions by more efficiently separating charge carriers [11–15]. Based on the band alignment fundamental, the rational design of the type II heterostructured photocatalysts had to consider the matched conduction and valence band positions. This means that the main challenge lies in eliminating the interfacial barrier of charge transfer between two solid-state absorbers, which is highly desirable.

Leaning from natural Z-scheme photosynthesis systems, where solar light is captured by the photosystem (PS I) consisting of a set of assembled complexes that smoothly funnel light energy into the photosystem (PS II) performing photochemical reactions, the concept of the Z-scheme photosystem has been utilized by Abe and coworkers for overall water splitting under visible light in 2001 [16]. They used two discrete narrow-bandgap semiconductors to construct an artificial Z-scheme photosystem in the presence of a suitable redox mediator [17,18], where the electrons produced on the $O_2$-evolving semiconductor were transferred via the mediator, to combine with the holes excited by the $H_2$-evoluting photocatalyst. The photogenerated electrons and holes were eventually placed in the two separated semiconductors with the higher net conduction (CB) and valence band (VB) potentials, respectively. Hence, the artificial Z-scheme photosystem could not only efficiently separate the photoinduced electrons and holes, but also maintain the redox potential of photointroduced electrons and holes in the separated semiconductors [19]. Different from the overall reaction of water splitting, wastewater purification requires more in the band structure of semiconducting photocatalysts. The pivotal requirement is that one of the two separated semiconductors has a VB potential larger than the oxidization potential of water to hydroxyl radicals ($\cdot OH/H_2O$ = +2.4 eV vs. NHE) and that another one possesses a CB potential more negative than the reduction potential of oxygen to superoxide radicals ($\cdot O_2^-/O_2$ = −0.33 eV vs. NHE) [20]. Among many well-studied semiconductors that can work for the hole oxidation of water into hydroxyl radicals, beside $TiO_2$ and ZnO, tungsten oxide ($WO_3$), with a bandgap energy of ca. 2.8 eV, is the ideal one because it has a more positive VB potential (ca. +3.1 eV vs. NHE) and can absorb visible photons [21–23]. For the other half, tin indium sulfide ($SnIn_4S_8$) is screened out to couple with $WO_3$, based on the following facts: (1) $SnIn_4S_8$ (SIS) is also an efficient visible light absorber with direct bandgap energy of ca. 2.0 eV. Its CB potential is positioned at −0.7 eV vs. NHE, which is more negative than the standard redox potential of $\cdot O_2^-/O_2$ [24]. (2) The VB potential of SIS is positioned at +1.30 eV vs. NHE, close to the CB position (+0.26 eV vs. NHE) of $WO_3$. The difference between the two semiconductors is equal to 1.37 eV. Direct integration of these into a type II heterostructured photocatalyst is unsuitable because the holes excited in $WO_3$ can oxidize SIS, resulting in the rapid photocorrosion. Therefore, seeking a suitable charge shuttle mediator to suppress photocorrosion is the more important task in this work, even though they are a couple of ideal candidates for the rational design of artificial Z-scheme photosystems for water purification containing Cr(VI) ions.

In general, for Z-schematic water photosplitting [25–27], there are two kinds of reversible charge mediators: redox ion pairs ($Fe^{3+}/Fe^{2+}$, $IO_3^{3-}/I^-$) [28–30] and noble metals (e.g. Ag, Au, Pt). However, the backward reactions easily occur with use of these noble metal mediators. Moreover, the solid-state mediators have one other drawback: The surface plasmon resonance greatly decreases the visible light harvesting of two semiconductors. From the viewpoints of cost and practical application, these noble metal mediators are undesirable for water purification. Owing to the easy self-redox decomposition and the possible introduction of other pollutants, the ionic redox mediators are not optimal for purification of water containing high-valence chromium ions. Reduced graphene oxide (rGO) is an alternative to solid-state charge mediators. It was first developed as an electron mediator by Amal and coworkers for splitting water in 2011 [31], and has recently received more attention. This attention has been paid mainly to its applications in construction of graphene-based photocatalysts, due to its low-cost, facile, and high-yield preparation [32–34]. Compared to the noble metal electron mediators, two-dimensional rGO nanosheets have more advantages. Besides

the high mobility of charge carriers [35], which facilitate the collection of photogenerated electrons and holes from the two photoexcited semiconductors, the larger specific surface area offers a better platform for collecting the reactive substrates and supporting the semiconductor nanoparticles [36]. Furthermore, the work function in the 5.0–6.7 eV range for rGO, highly comparable to noble metals, is higher than those of SIS (4.01 eV) and $WO_3$ (4.86 eV) [37,38]. This strongly encourages us to modulate the carrier behaviors of SIS and $WO_3$ with rGO. The main objective of this study is to construct an all-solid-state Z-schematic $WO_3/rGO/SnIn_4S_8$ (WGS) photocatalyst for efficient water purification under visible light irradiation, with rGO as the build framework for shuttling charge carriers and bridging the two visible light absorbers, SIS and $WO_3$. Compared with other Z-scheme heterojunctions, graphene-based Z-scheme heterojunctions exhibit many advantages, such as easy fabrication, tunable band structures, and increased specific surface area and adsorption sites [39]. Over the past few years, great efforts have been made in the fields of energy conversion, organic synthesis, and wastewater treatment [40–42]. Dye and Cr(VI), which were widely utilized in many industries (e.g., textile manufacturing, leather tanning, and electroplating), have been studied as target pollutants to investigate the photocatalytic activity of photocatalysts. Wang et al. [43] constructed a novel Z-scheme photocatalyst with rGO as the electron mediator for enhanced photocatalytic degradation of RhB, where the removal efficiency increased from 74.5% to 92.3%. Chen et al. [44] fabricated an all-solid-state Z-scheme heterojunction $BiOI/rGO/Bi_2S_3$ to simultaneously remove of Cr(VI) and organic pollutant, in which the optimum reductive and oxidative efficiencies were up to 73% and 95%. Obviously, graphene-based heterojunction photocatalysts hold great promise for water purification.

Herein, a combined protocol of in situ precipitation method with hydrothermal synthesis was developed to realize the perfect integration of the two semiconductors with the charge mediator to form a "sandwich" nanohybrid, as depicted in Scheme 1. Thin $WO_3$ nanosheets were deposited in advance on the graphene oxide (GO) surface, and metal ions, $In^{3+}$ and $Sn^{4+}$, were added into the $GO/WO_3$ solution, where these metal ions can be adsorbed on the GO surface by the electrostatic interaction with the negative carboxylate acid (–COO$^-$) and hydroxide (–OH) groups. Then, sodium diethyldithiocarbamate (DDTC-2Na) was added into the precursor solution as a sulfur source, to grow SIS nanosheets onto the GO surface in situ and to prevent GO agglomeration during the hydrothermal synthesis. Typically, the convenient hydrothermal reaction was carried out at 180 °C, after reaction of 12 h. GO can be reduced to rGO by removing carboxylate acid groups and finally the $WO_3/rGO/SnIn_4S_8$ sandwich hybrids were produced. The crystal phase, morphology, and chemical composition of the as-prepared resultants were characterized by XRD, SEM, HRTEM, UV-vis DRS, PL, and ESR analysis. Meanwhile, the photocatalytic performance was evaluated by two model reactions, RhB degradation and Cr(VI) reduction, under visible light irradiation. It was found that the hybrid photocatalyst exhibited enhanced photocatalytic activity. Moreover, a Z-scheme mechanism for the enhanced photocatalytic activities was proposed based on the reactive oxygen species trapping experiments and the ESR characterizations. This work will provide a novel two-dimensional Z-scheme system for efficient degradation of pollutant, and has potential application in both environmental protection and energy conversion.

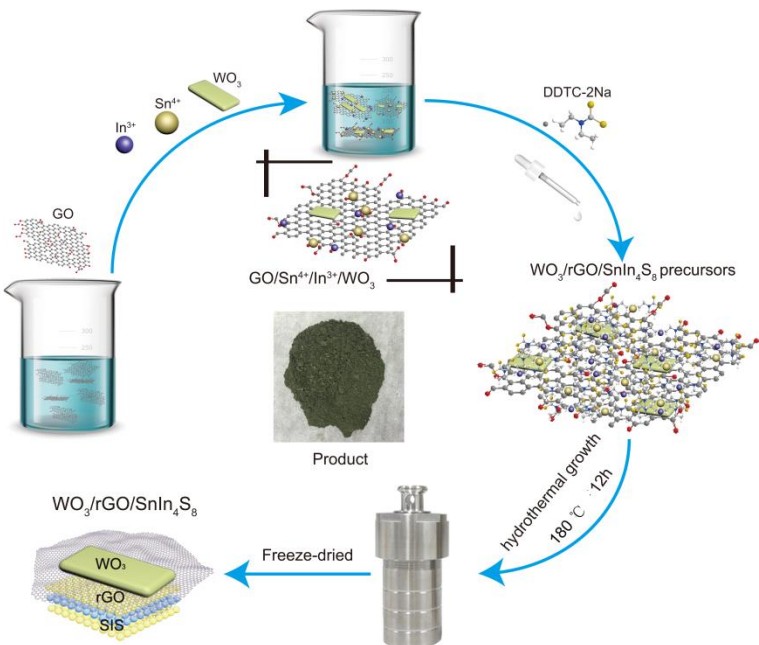

**Scheme 1.** Schematic illustration of the synthetic procedure of $WO_3/rGO/SnIn_4S_8$ (WGS) heterojunction.

## 2. Results and Discussion

### 2.1. Structural Characterization of WGS Photocatalysts

The crystal structure and phase composition of the as-prepared samples were characterized by XRD spectroscopy (Figure 1). The pristine SIS displayed the diffraction peaks at $2\theta = 14.3°, 27.5°, 33.3°, 43.7°$, and $47.8°$ which well match the (111), (311), (400), (333), and (440) crystal planes of cubic SIS (JCPDS card # 42-1305, a = b = c = 10.7507 Å). The result is in agreement with our previous report [24]. The predominant diffraction peaks of pure $WO_3$ at $2\theta = 23.1°, 24.4°$, and $34.2°$, attributed to (002), (020), (200), and (202) crystal planes of monoclinic $WO_3$ (JCPDS card # 43-1035, a = 7.297 Å, b= 7.539 Å, c = 7.688 Å). In general, the XRD patterns of WGS hybrids are similar to that of the pure SIS. Compared with the diffraction peaks of SIS, the peak intensity of pristine $WO_3$ is relatively weak due to the low dosage of $WO_3$ in the WGS composites. Besides, a typical peak around $24.6°$ in the pattern of WGS hybrids belongs to the (100) plane diffraction pattern of rGO. No diffraction peaks of other impurities were detected, indicating that WGS hybrids were successfully prepared. It is also worth mentioning that none of the diffraction peaks of $WO_3$ were shifted to larger angles after integration with SIS via hydrothermal process, thus indicating that sulfide-doping did not occur on W or O sites.

The morphology of the as-synthesized samples was examined by SEM and TEM. Both the SEM images of rGO and SIS exhibit wrinkle, ultrathin, and layered structures, while the original $WO_3$ is an urchin-like hierarchical structure, which is similar to the previous report (seen in Figure S1, ESI†) [45]. Typically, a small and thin photocatalyst nanosheet is suitable for the construction of the sandwich structure. In our case, the hierarchical structure of $WO_3$ was further heated at 600 °C to obtain a regular nanosheet with a thickness in the range of 14–57 nm and an average size of 40 nm. As shown in Figure 2A, the SEM image of WGS-5% hybrids show the in situ growth of SIS onto rGO to form a flexible structure, indicating that the SIS nanosheet is well dispersed on the surfaces of the rGO sheet at high densities. The small flake placed between the rGO and SIS surface may be assigned to the $WO_3$ nanosheet, whose size is smaller than the rGO and SIS nanosheet.

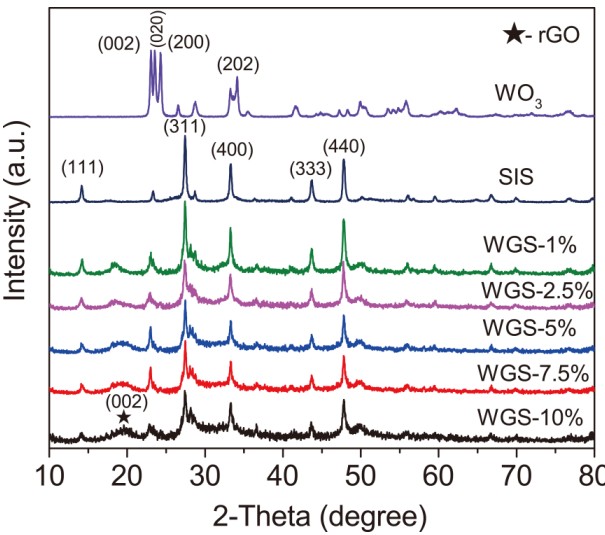

**Figure 1.** XRD patterns of $WO_3$, $SnIn_4S_8$ (SIS), and WGS photocatalysts.

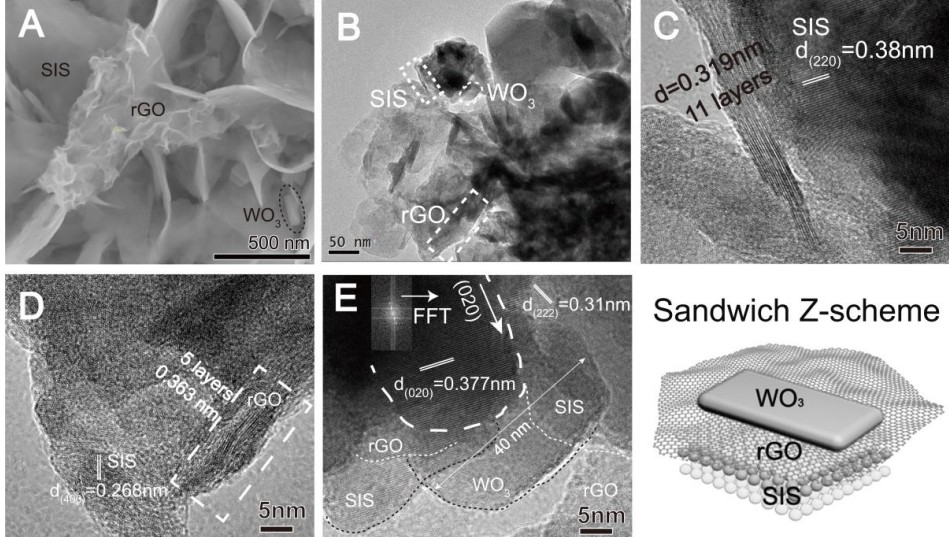

**Figure 2.** (**A**) SEM images of WGS-5%; (**B**) TEM images and (**C–E**) HRTEM images of WGS-5%.

HRTEM image of WGS composites was observed to further reveal the morphology and crystal structure. As shown in Figure 2B,E, SIS and $WO_3$ uniformly pave on the surface of rGO to form a sandwich structure, indicating the artificial Z-scheme photocatalyst was successfully synthesized. It is worth noting that the large specific area of rGO is suitable for efficiently separating the photoinduced electrons and holes. Moreover, the lattice fringe was investigated to reveal the crystal information of $WO_3$, SIS, and rGO. As shown in Figure 2C–E, the lattice spacings of 0.310 nm, 0.38 nm, and 0.268 nm can be respectively assigned to the (222), (220), and (400) planes of SIS, which agrees with the result of XRD analysis. The lattice spacing (0.377 nm) and corresponding FFT pattern in Figure 2E confirm the monoclinic $WO_3$ deposit on the rGO sheet. Furthermore, Figure 2B,C show that the thicknesses of the SIS and rGO sheets are 10–12 layers and 5–7 layers, respectively, indicating that the GO was not agglomerated via hydrothermal reduction. The obvious sandwich structure demonstrated that the Z-scheme heterostructure was successfully fabricated by a combined protocol of the in situ precipitation method with hydrothermal synthesis.

The compositional information and chemical states of WGS hybrids were investigated by XPS analysis. As shown in Figure 3A, the survey scan of WGS-5% hybrid demonstrates the existence of C, O, S, In, and W without other impurity elements. The high-resolution C1s spectrum shown in

Figure 3B can be resolved into three peaks centered at 284.8, 286.0, and 289.4 eV, which can be assigned to the $sp^2$ C=C bond, $sp^3$ C–O bond, and $sp^2$ C=O bond, respectively. The peak area ratios of the C–O, C=O to the C=C bond were calculated, shown in Figure 3B (inset table). It is obvious that the percentage of these oxygen-containing functional groups in the sample decreases to a large degree compared to that in the pure GO (Figure S2, ESI†), indicating that GO was reduced to rGO with a small amount of residual oxygen-containing groups via hydrothermal reaction.

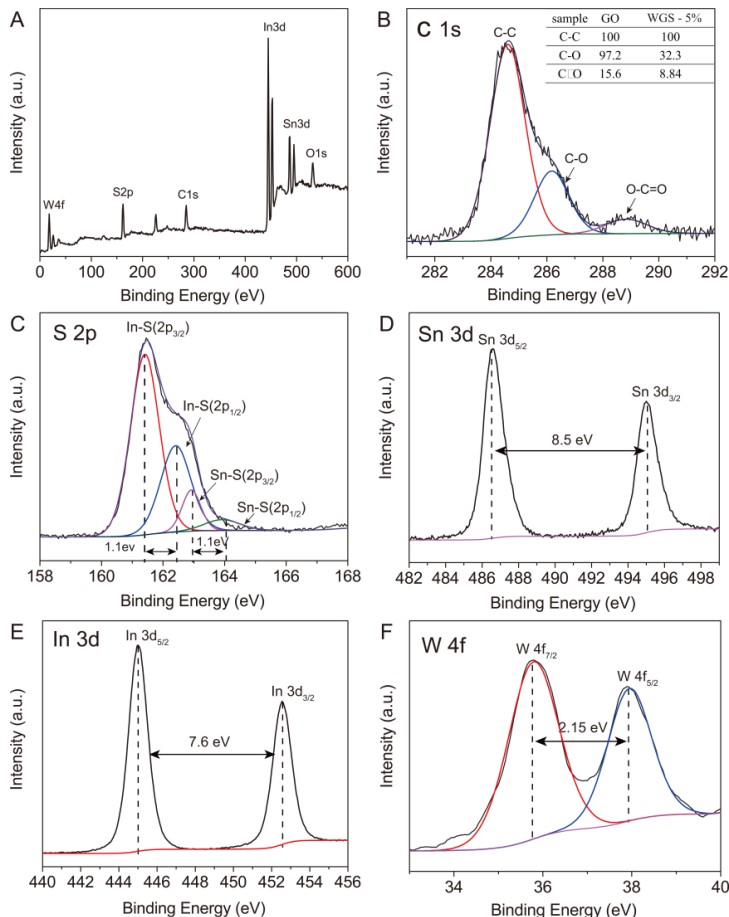

**Figure 3.** XPS spectra of WGS-5% photocatalyst: (**A**) Survey scans spectra; (**B**) C 1s; (**C**) S 2p; (**D**) Sn 3d; (**E**) In 3d; and (**F**) W 4f.

Regarding the S 2p spectrum, two group peaks are assigned to the In–S and Sn–S bonds. The detail scans of S 2p peaks show a broad spectrum with a hump at higher binding energy. As previously reported, the S $2p_{1/2}$ peak of semiconductor chalcogenide appears at higher binding energy ($1.2 \pm 0.1$ eV) than the S $2p_{3/2}$ peak [46]. In this case, the S 2p peak divide into two groups, which are respectively assigned to the In–S and Sn–S bonds. The group of In–S $2p_{3/2}$ and In–S $2p_{1/2}$ peaks appear at 161.4 and 162.4 eV, respectively, whereas the Sn–$S_{3/2}$ and Sn–$S_{1/2}$ center at 162.9 eV and 164.0 eV, respectively. Moreover, the peak area ratios and the spin orbital splitting were calculated (Table S1, ESI†) to identify the chemical states of In 3d, Sn 3d, and W 4f [47], indicating that the valence states in the composite are $In^{3+}$, $Sn^{4+}$, and $W^{6+}$, respectively.

The bandgap of the as-prepared sample was estimated from UV-vis DRS spectra via Tauc's plot. Equation:

$$(\alpha h v)^n = A(h v - E_g) \tag{1}$$

where $\alpha$, $hv$, A, and $E_g$ are the absorption coefficient, incident photon energy, constant, and band gap energy, respectively. The value of $n$ is related to the transition feature of electrons in a semiconductor (indirect transition: $n = 1/2$; direct transition: $n = 2$). As shown in Figure 4A, all the samples exhibit an

absorption edge in the visible light region. Pure SIS and WO₃ show an absorption edge at ~635 nm and ~465 nm, corresponding to bandgaps of 1.95 and 2.67 eV, respectively.

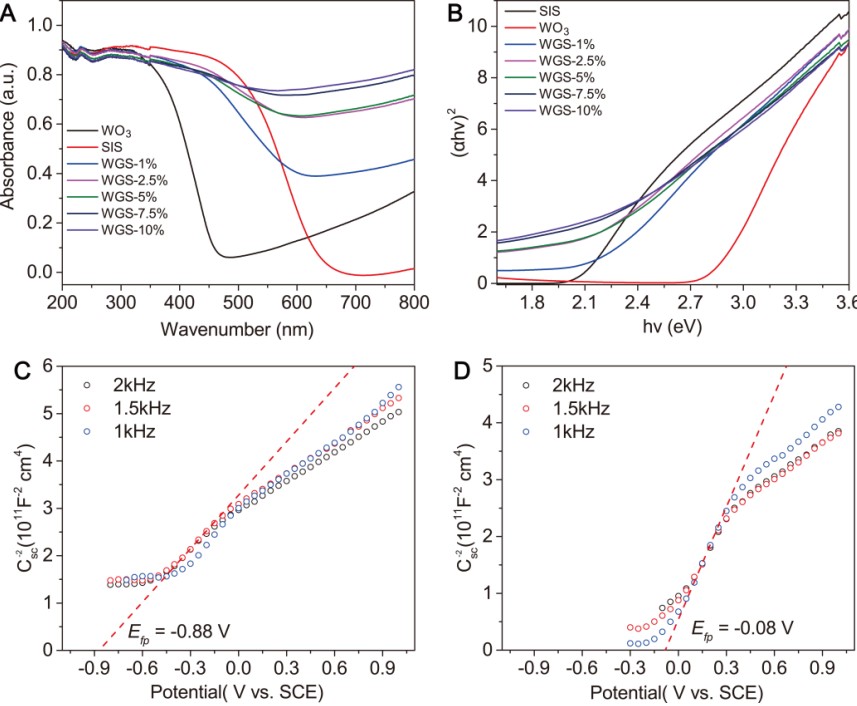

**Figure 4.** (**A**) UV-vis spectra of WO₃, SIS, and WGS photocatalysts; (**B**) Plots of (*αhv*)² *vs* (*hv*) for WO₃, SIS, and WGS photocatalysts; Mott–Schottky plots of (**C**) SIS and (**D**) WO₃.

For the WGS composites, there is a parabolic relationship between rGO content and the absorption edge, while the mass ratio of SIS to WO₃ is kept constant. Increasing GO content from 0 to 10 percent, the absorption edge shifts from ~563 nm (2.2 eV, 1 wt%) to ~605 nm (2.05 eV, 5 wt%), and then decreases to ~576 nm (2.15 eV, 10 wt%). It should be noted that the addition of rGO decreases the bandgap due to its strong visible light absorption ability [48]. However, excessive rGO decreases the visible light absorbance, possibly because excessive rGO prevents light from reaching SIS. Therefore, all the samples exhibit a broad background in the visible light region.

The flat-band potential of WO₃ and SIS were measured by an electrochemical method and calculated by the Mott–Schottky equation [49]:

$$\frac{1}{C_{sc}^2} = \frac{2}{\varepsilon\varepsilon_0 e N_D A^2}\left(V - E_{fb} - \frac{k_B T}{e}\right) \tag{2}$$

where $C_{sc}$ is the space charge capacitance, $\varepsilon$ is the dielectric constant of the semiconductor, $\varepsilon_0$ is the vacuum permittivity ($8.85 \times 10^{-12}$), $N_d$ denotes the donor density, $E$ is the applied potential, $E_{fb}$ is the flat-band potential, $T$ is the absolute temperature, $k_B$ is Boltzmann's constant ($1.38 \times 10^{-23}$ J·K⁻¹), and $V$ is the applied voltage. As shown in Figure 4C,D, the slope of Mott–Schottky plots is positive, which demonstrates that SIS and WO₃ were *n*-type semiconductors [50]. Plot of $1/C_{sc}^2$ against $V$ should yield a straight line from the curve of Mott–Schottky ($1/C_{sc}^2 = 0$). The $E_{fb}$ can be calculated by the equation of $E_{fb} = V - k_B T/e$, listed in Table S2. The $E_{fb}$ values of SIS and WO₃ are $-0.90$ and $-0.13$ V versus the saturated calomel electrode (SCE), respectively [51].

### 2.2. Photocatalytic Performance

The photocatalytic activities were examined by the degradation of RhB and reduction of Cr(VI) under visible light irradiation. Prior to irradiation, the samples were stirred in the dark for 60 min to

ensure the adsorption–desorption equilibrium. As shown in Figure 5A, the nanohybrids of WGS-1%, WGS-2.5%, and WGS-5% show enhanced photocatalytic activity on degradation of RhB in comparison with pure SIS and WO$_3$. In the presence of SIS and WO$_3$ catalysts, 88% and 23% RhB degradation efficiency, respectively, was achieved after 60 min visible light irradiation. The low photocatalytic efficiency of WO$_3$ may be ascribed to the fast recombination of the photoinduced hole and electron. The maximum RhB degradation efficiency was achieved with 5% rGO loading, however, excessive loading leads to the decline of the photocatalytic efficiency. Similar results were reported with previous graphene-based photocatalysts, indicating that rGO loading has a significant influence on the photocatalytic activity in Z-scheme hybrids [52,53]. The RhB degradation rate constants are in the order of WGS-5% > WGS-2.5% > WGS-1% > SIS > WGS-7.5% > WGS-10% > WO$_3$ (Figure 5B), where WGS-5% shows the highest value of $k_{app}$ = 0.085 mg·L$^{-1}$·min$^{-1}$, which is ca. 2.4 and 20 times more than those of pure SIS and WO$_3$, respectively. To further investigate the reduced ability of the as-synthesized sample, photocatalytic reduction of Cr(VI) was carried out for comparison. As shown in Figure 5D, the optimized case of the WGS-2.5% photocatalyst with a broad photo-response exhibits the highest Cr(VI) reduction rate, which is ca. 1.8 and 12 times more than those of pure SIS and WO$_3$. Obviously, the integration of rGO could enhance the photocatalytic performance due to its high mobility of charge carriers [54]. However, excessive rGO leads to the decrease of the degradation and reduction rate, possibly because the agglomeration of rGO covered on the surface would prevent visible light from reaching the surface of the photocatalyst. As a result, a suitable percentage of rGO loading is crucial for the enhancement of photocatalytic activity. Previous research has demonstrated that rGO can efficiently separate the charge carriers between two semiconductors to enhance the photocatalytic degradation/reduction. Compared to a previous report of another graphene-based Z-scheme system (Table S3, ESI†), the present work exhibits remarkable photocatalytic activity of RhB and Cr(VI). In addition, the stability and reusability of WGS-2.5% composite has been assessed by recycled experiment and XRD analysis. The removal efficiency of Cr(VI) remained at 95.4% after four cycles and the diffraction peak intensity of XRD pattern remained unchanged (Figure S4) compared to the fresh sample, which exhibited the excellent stability of this kind of photocatalyst.

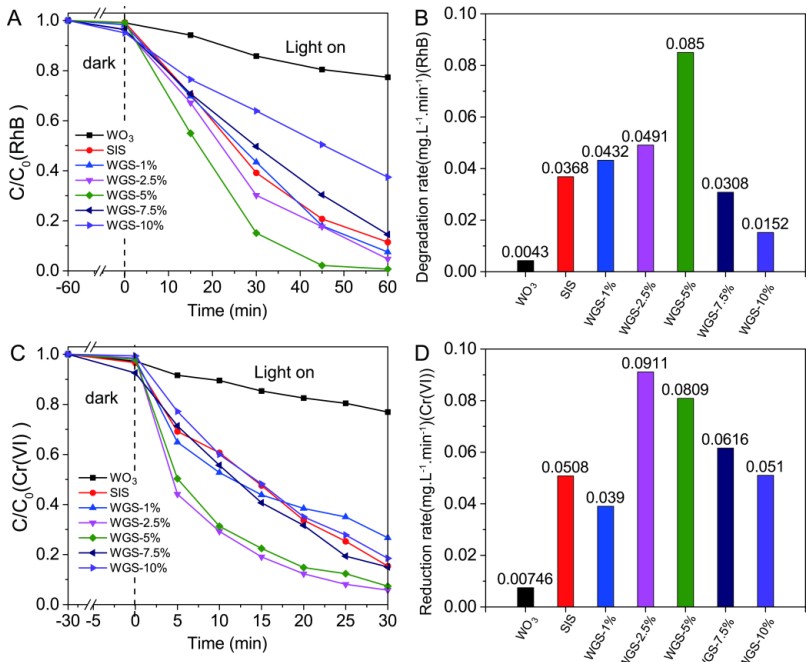

**Figure 5.** (**A**) Photocatalytic degradation of RhB in the presence of WO$_3$, SIS, and WGS photocatalysts; (**B**) degradation rate of RhB; (**C**) reduction of Cr(VI) in the presence of WO$_3$, SIS, and WGS photocatalysts; (**D**) reduction rate of Cr(VI).

## 2.3. Mechanism of the Photocatalytic Activity

Photocatalytic activity is closely related to the type and amount of the reactive oxygen species originating from the photogenerated electron and holes. In general, $\cdot OH$, $\cdot O_2^-$, and $h^+$ are the major reactive species responsible for the photo-oxidation of environmental pollutants under visible light irradiation. To demonstrate the molecular mechanism of RhB degradation, 1,4-benzoquinone (BQ), isopropanol (IPA) and ammonium oxalate (AO) were used for trapping the radical species of $\cdot O_2^-$, $\cdot OH$, and photogenerated holes ($h^+$), respectively. As shown in Figure 6A, the RhB degradation efficiency over the WGS-5% is significantly reduced with the addition of trapping agent. It is observed that only 68%, 77%, and 94% of degradation efficiency in the presence of BQ, IPA, and AO, respectively, compared to the degradation efficiency of blank reaction, which is up to 99% without adding scavenger. The effect of the reactive species in photodegradation of RhB is $\cdot O_2^- > \cdot OH > h^+$, suggesting that $\cdot O_2^-$ plays a dominant role in the photocatalytic reaction process. To further quantify the yield of $\cdot OH$ in the photocatalysis solution of the pure and composite samples, TA was used as a trapping agent to react with hydroxyl radicals to form a fluorescent active species (2-hydroxyterephtalic) [55]. As shown in Figure 6C, the fluorescence intensity of WGS-5% is apparently higher than that of the pure SIS and $WO_3$ under visible light irradiation, indicating that the Z-scheme hybrids have a stronger VB potential to oxidize water to hydroxyl radicals.

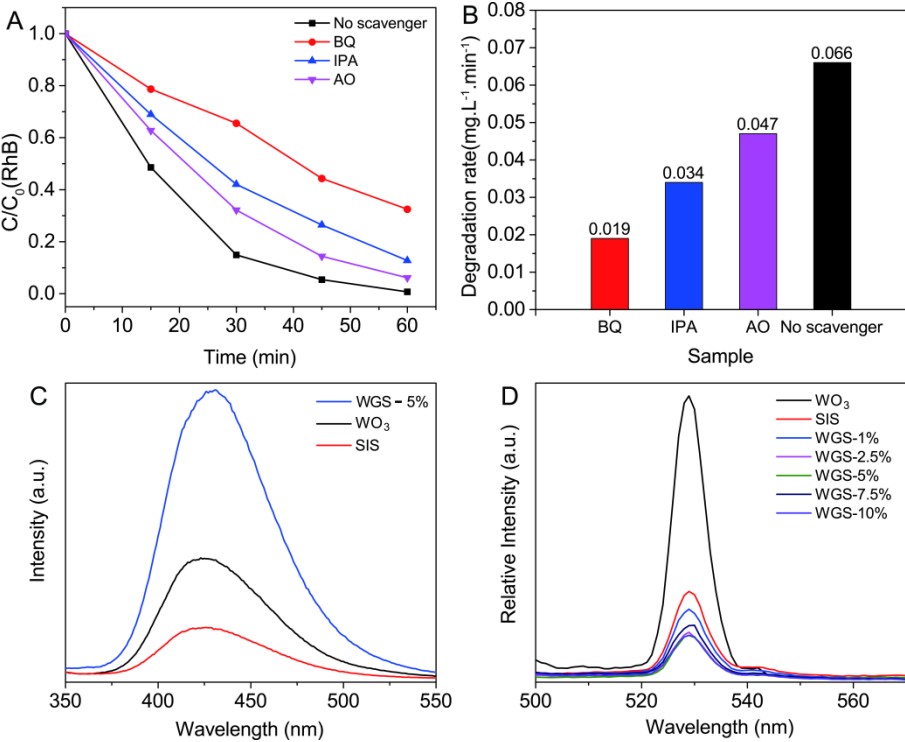

**Figure 6.** Effect of scavengers on RhB degradation over WGS-5%: (**A**) Photocatalytic efficiency and (**B**) degradation rate; (**C**) PL spectra of TA in the presence of $WO_3$, SIS, and WGS-5% photocatalysts; (**D**) PL emission spectra ($\lambda_{ex}$ = 350 nm) of $WO_3$, SIS, and WGS photocatalysts.

ESR analysis was used to further evidence the reactive oxygen species of $\cdot O_2^-$ and $\cdot OH$ in the photocatalytic systems of $WO_3$, SIS, and WGS-5% under visible light. As shown in Figure 7A, six characteristic peaks of the DMPO-$\cdot O_2^-$ adduct and four characteristic peaks of the DMPO-$\cdot OH$ can be observed for all the samples, which indicates that $\cdot O_2^-$ and $\cdot OH$ species are generated in the photocatalytic reaction. The signal intensities of both DMPO-$\cdot O_2^-$ and DMPO-$\cdot OH$ of the WGS hybrids are obviously stronger compared to pure SIS and $WO_3$, suggesting that the Z-scheme photocatalyst could efficiently separate the photoinduced electron–hole pairs and thus accelerate the photocatalytic activity. This result is also in agreement with the above active species trapping

experiment. Moreover, the signal intensity of DMPO-·$O_2^-$ generated by SIS was stronger than WO$_3$, while the signal DMPO-·OH was weaker. This indicates that ·OH are the main reactive species of pure WO3 in the photocatalytic reaction, and further indicates that both the ·$O_2^-$ and ·OH play a major role in the photocatalytic process for WGS hybrids.

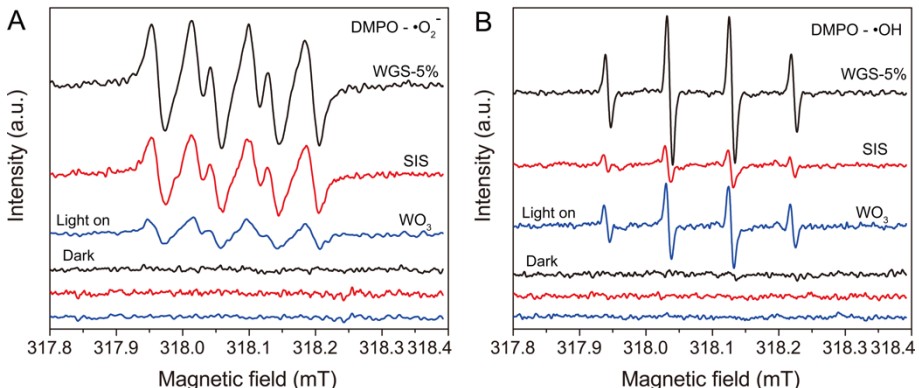

**Figure 7.** ESR signals of the (**A**) DMPO-·$O_2^-$ and (**B**) DMPO-·OH of WO$_3$, SIS, and WGS-5% photocatalysts.

Photoluminescence (PL) spectroscopy was employed to investigate the quantum efficiency of WGS Z-scheme. Generally, weaker PL spectrums mean higher separation rates of photoinduced charge carriers, and possibly higher photocatalytic activity [56]. Due to the integration of the charge mediator of rGO, the PL intensity of the WGS hybrids is weaker than those of pure SIS and WO$_3$ (Figure 6D), which demonstrates the higher separate efficiency of electron–hole pairs.

In order to reveal the enhanced photocatalytic activity of Z-scheme mechanism, the optical structure of SIS and WO$_3$ was confirmed and the possibility of electron transfer pathways was presented. According the bandgap level analysis, both pure WO$_3$ and SIS ha d narrow-bandgap energies of 1.95 and 2.67 eV, which demonstrated excellent visible light property. The CB potential of SIS is positioned at −0.66 eV (vs. NHE), while the O$_2$-evoluting photocatalyst WO$_3$ has a positive VB potential of 2.79 eV (vs. NHE). As a result, the electrons generated on the CB of SIS can reduce O$_2$ into ·$O_2^-$ radicals since the position of CB is more negative than the potential of the ·$O_2^-$/O$_2$ (−0.33 eV vs. NHE), whereas the VB potential of WO$_3$ is lower than the standard redox potential of ·OH /H$_2$O (2.4 eV vs. NHE), implying that the photoexcited holes of WO$_3$ can oxidize the absorbed H$_2$O molecules to produce ·OH. The major reaction steps in this mechanism under visible light irradiation are summarized as following:

$$\text{WGS} \xrightarrow{h\nu} \text{SIS}\left(e^-/h^+\right)/\text{rGO}/\text{WO}_3\left(e^-/h^+\right) \tag{3}$$

$$\text{SIS}\left(e^-/h^+\right)/\text{rGO}/\text{WO}_3\left(e^-/h^+\right) \xrightarrow{Z-scheme} \text{SIS}\left(e^-\right)\text{rGO}\left(e^-_{\text{SIS}}/h^+_{\text{WO}_3}\right) + \text{WO}_3\left(h^+\right) \tag{4}$$

$$\text{SIS}\left(e^-\right) + \text{O}_2 \rightarrow \text{O}_2^- \tag{5}$$

$$\text{WO}_3\left(h^+\right) + \text{H}_2\text{O} \rightarrow \cdot\text{OH} \tag{6}$$

$$6\text{SIS}\left(e^-\right) + \text{Cr}_2\text{O}_7^{2-} + 14\text{H}^+ \rightarrow 2\text{Cr}^{3+} + 7\text{H}_2\text{O}. \tag{7}$$

$$\cdot\text{O}_2^-/\cdot\text{OH} + \text{RhB} \rightarrow \cdots \rightarrow \text{CO}_2 + \text{H}_2\text{O} \tag{8}$$

Based on the results above, we proposed a Z-scheme mechanism for enhanced photocatalytic activity with rGO as the electron mediator. As illustrated in Figure 8, both SIS and WO$_3$ can be excited by visible light irradiation and have a suitable redox potential to form the reactive oxygen species. Upon irradiation, photogenerated electrons in the CB of WO$_3$ transfer via rGO to combine with the holes in VB of SIS, resulting in the accumulation of electrons in the CB of SIS and holes in the VB

of $WO_3$. The separation of photointroduced charge carriers in space prolonged their lifetime and increased the probability of the photoexcited electron and hole participating in the photocatalytic reaction, which will significantly enhance the photocatalytic activity. Since the CB position of SIS is more negative than the $\cdot O_2^-/O_2$ potential, the electrons accumulated in the CB of SIS can easily reduce the adsorbed $O_2$ to produce $\cdot O_2^-$, which is well proved by the trapping and ESR experiment. It is worth noting that the composite produced higher $\cdot OH$ concentrations than pure $WO_3$ due to the lower recombination rate of electrons and holes. This result is further demonstrated by the PL analysis. Therefore, it can be concluded that the photocatalytic reaction of the WGS heterojunction followed a solid-state Z-scheme mechanism, which could improve the separation of the electron–hole pairs and exhibit a strong redox ability for efficient degradation of organic pollutants and reduction of Cr(VI).

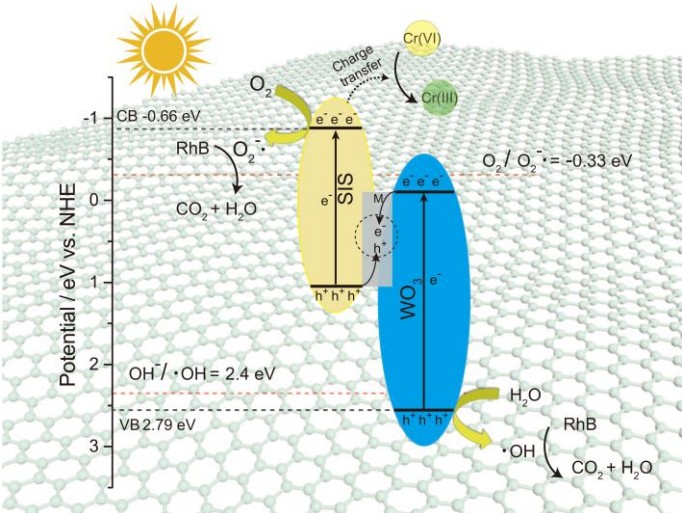

**Figure 8.** Schematic illustrations of a Z-scheme mechanism with reduced graphene oxide (rGO) as the solid-state electron mediator.

## 3. Materials and Methods

### 3.1. Materials

Indium (III) trichloride ($InCl_3 \cdot 4H_2O$), tin (IV) chloride pentahydrate ($SnCl_4 \cdot 5H_2O$), tungsten chloride ($WCl_6$), and sodium diethyldithiocarbamate trihydrate (DDTC-2Na) were purchased from Aladdin Chemical Reagent Company, China. Natural graphite flakes with an average particle size of 100 meshes, potassium dichromate, and phosphorus pentoxide were purchased from Sinopharm Chemical Reagent Company, China. All the chemicals were of analytical grade and used as received without further purification.

### 3.2. Synthesis of Photocatalysts

GO was synthesized by the modified Hummers' method [57]. Urchin-like $WO_3$ was prepared by the previously reported method [45]. In order to obtain nanosheet structure, we further heated to decompose the hierarchical structure of the as-prepared $WO_3$. In a typical synthesis, 2 g of $WCl_6$ was added into 60 mL of absolute ethanol, and then loaded into a Teflon-lined autoclave and kept at a constant temperature of 180 °C for 12 h. Thereafter, the blue powder was washed with distilled water several times and dried at 60 °C for further use. Furthermore, the as-prepared $WO_3$ nanostructure was heated at 600 °C for 2 h with a rate of 25 °C·min$^{-1}$. The color of $WO_3$ changed from blue to yellow.

Z-schemed WGS sandwich nanohybrids were synthesized using a combined protocol of the in situ precipitation with hydrothermal synthesis method. In a typical synthesis procedure, a certain amount of GO and $WO_3$ nanosheets were dispersed respectively in deionized water followed by ultrasonication for 10 min. The above suspensions were stirred for 2 h to obtain a homogeneous liquid.

Next, $SnCl_4 \cdot 5H_2O$ (0.3 mM, dissolved in acetic acid) and $InCl_3 \cdot 4H_2O$ (0.3 mM, dissolved in acetic acid) taken into 1:4 molar ratio were mixed with $GO/WO_3$ composites. Finally, DDTC-2Na (0.24 mM, dissolved in deionized water) was added dropwise into the mixed solution and vigorously stirred for 2 h. The suspension was sealed in a 100 mL Teflon-lined autoclave and the reaction temperature maintained at 180 °C for 12 h. The mixture was collected by filtration after cooling down, and washed repeatedly with deionized water and anhydrous ethanol. The final products were freeze-dried to obtain the WGS nanosheet. Pure rGO, $WO_3$, and SIS were also prepared under the same conditions. For the WGS hybrids, the mass ratio of $WO_3$ to SIS was fixed at 1:10, and the percentage of GO to SIS ranged from 1% to 10%. The as-prepared composites were labelled as WGS-1%, WGS-2.5%, WGS-5%, WGS-7.5%, and WGS-10%, respectively.

### 3.3. Characterization

The crystalline structure of the samples were analyzed by a powder X-ray diffractometer (XRD, Miniflex600, Rigaku, Japan) by Ni-filtered Cu K$\alpha$ irradiation ($\lambda$ = 1.5406 Å) in the region of 10° to 80° with a scanning rate of 2°·min$^{-1}$. The morphology and microstructure of the samples were observed using a field emission scanning electron microscopy (FESEM, Nova Nano SEM450, FEI, America, operated at 20 kV) and a high resolution transmission electronic micrograph (HRTEM, Tecnai G220, FEI, America, operated at 200 kV). The chemical components and valence of the photocatalyst were measured by X-ray photoelectron spectroscopy (XPS, ESCA Lab250, Thermo Scientific Ltd, America) with Al K$\alpha$ radiation in twin anode, where the binding energies were calibrated by referencing C 1s (284.6 eV) peak. UV-vis diffuse reflectance spectra (DRS) were performed on a UV-vis spectrophotometer (Cary 5000, Agilent, America). ESR experiments were conducted with ESR spectrometer (JES-FA200, JESO, Japan). PL spectroscopy was examined on a fluorescence spectrophotometer (Cary Eclipse, Agilent, America).

### 3.4. Photocatalytic Testing

All the experiments were performed in a double-walled quartz jacket filled with cool water. A 300 W Xenon lamp (Beijing Aulight Ltd., China) coupled with a cutoff filter ($\lambda$ > 420 nm) was used as the light source, and the incident light intensity was 100 mW·cm$^{-2}$. The photocatalytic activity of the samples were evaluated by the degradation of RhB and reduction of Cr(VI) under visible light irradiation. In brief, 10 mg of photocatalyst was added to 100 mL of RhB or Cr(VI) solution with initial concentration ($C_0$) of 30 and 20 mg·L$^{-1}$, respectively. Prior to irradiation, the suspension was magnetically stirred in dark conditions for 30 min to reach the adsorption–desorption equilibrium. Then, the above suspension was illuminated by Xe lamp with magnetic stirring and 3 mL of suspension was sampled at given time intervals. The suspension was centrifuged to separate the residual photocatalysts from the solution. The concentration of Cr(VI) was determined by the diphenylcarbazide colorimetric method. The residual RhB and Cr(VI) concentration was measured by a UV-vis spectrophotometer (Shimadzu, Uv-2450) at the maximum absorption wavelengths of 554 nm and 540 nm, respectively [58]. Finally, the photocatalytic efficiency was calculated according to $C/C_0$, where $C_0$ and $C$ represent the concentration of RhB and Cr(VI) before visible light irradiation and after reaction.

### 3.5. Radical Species Trapping and ESR Experiments

To investigate the major reactive oxygen species in photocatalytic reaction, 1,4-benzoquinone (BQ 0.1 mM), isopropanol (IPA, 1 mM), and ammonium oxalate (AO, 1 mM) were selected as scavengers for trapping superoxide radicals ($\cdot O_2^-$), hydroxyl radicals ($\cdot OH$), and photogenerated holes ($h^+$), respectively. The experimental conditions were similar to the above photocatalytic activity test. We further quantitated the species of $\cdot OH$ by PL spectroscopy using terephthalic acid (TA) as the fluorescent scavenger. Furthermore, the molecular mechanism of the enhanced photocatalysis was revealed by ESR experiments. In the typical procedure, 5 mg of the as-prepared samples was dispersed

in 0.5 mL deionized water or methanol, and then 45 μL of DMPO was added followed by ultrasonic for 5 min. All activity species trapping experiments were carried out under visible light irradiation (λ > 420 nm).

## 4. Conclusions

In summary, a novel all-solid-state Z-scheme photocatalyst has been successfully prepared via a combined protocol of the in situ precipitation method with hydrothermal synthesis. It features a sandwich heterostructure, where rGO bridges $SnIn_4S_8$ and $WO_3$ nanosheets. The sandwich nanohybrids display higher photocatalytic activities for both of the degradation of RhB and reduction of Cr(VI), compared to the pure $SnIn_4S_8$ and $WO_3$. The characterization results indicate that the Z-scheme mechanism enables the efficient separation of photogenerated charge carriers by use of rGO as the solid-state charge mediator. This work contributes to a novel strategy for the design and preparation of Z-scheme photocatalysts for highly efficient elimination of environmental pollutants.

**Supplementary Materials:** The following are available online at http://www.mdpi.com/2073-4344/9/2/187/s1. Figure S1: SEM images (A) rGO; (B) SIS; (C) original WO3; (D) WO3, Figure S2: XPS spectra of GO; detail scan of C 1s, Table S1: peak separation and area ratios of Sn, In, and W.

**Author Contributions:** Conceptualization, M.L. and H.F.; Methodology, P.X., M.L. and H.F.; Validation, M.L. and H.F.; Formal Analysis, P.X.; Investigation, P.X., Z.W. and S.H.; Resources, M.L. and H.F.; Data curation, Y.L., J.L. and W.Z.; Writing—Original Draft Preparation, P.X.; Writing—Review & Editing, Y.L., J.L., W.Z. and H.F.; Supervision, M.L. and H.F.; Administration, H.F.

**Funding:** This work was financially supported by the Natural Science Foundation of China (Grants No. 21773031 and 21577018), National Key Research and Development Program of China (2017YFB0308600).

**Acknowledgments:** We greatly thank Natural Science Foundation of China and National Key Research and Development Program of China for providing funding to carry out this research. We are also pleased to acknowledge Fuzhou University for providing the necessary facilities to carry out this work.

**Conflicts of Interest:** The authors declare no conflict of interest.

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
