# Peer review of "Z-Schemed WO3/rGO/SnIn4S8 Sandwich Nanohybrids for Efficient Visible Light Photocatalytic Water Purification"

_catalysts, doi:10.3390/catal9020187_

Round 1
Reviewer 1 Report
The authors have well discussed the photocatalytic activity through their synthesized photocatalyst. It deserves to be published since the paper is attractive!
However, it needs a minor revision before publishing:
-Reference is not unified and needs modification. For instance, take a look at ref. 18, 44, 45, 34,
In ref 28, the Authors name have been wrongly written.
-In the introduction, the authors need to discuss about the exfoliation of graphene based materials and a brief background about its catalytic updates. Authors can use following references: ChemCatChem, 2015, 7, 1678-1683; J. Coll. Inter. Sci. 2016, 478, 280-287; Int. J. Biol. Macromol. 101, 696–702.
-I think the SEM image shows that WO3 is bigger than rGO, while the authors in their schemes have mentioned that rGO. So I think authors should reconsider this matter and give a proof of this claim. I think it is better to measure the particle size of WO3 through TEM image. And please bring a understandable image of SEM.
-English also needs to be polished.
After above revision, it can be published.
Author Response
Reviewer #1: The authors have well discussed the photocatalytic activity through their synthesized photocatalyst. It deserves to be published since the paper is attractive! However, it needs a minor revision before publishing.
1. Reference is not unified and needs modification. For instance, take a look at ref. 18, 44, 45, 34, In ref 28, the Authors name have been wrongly written.
Response:We are really sorry about this mistake that we could have avoided. Accordingly, we have re-checked throughout the whole references and revised the format to meet the demand of Catalysts Journal.
2. In the introduction, the authors need to discuss about the exfoliation of graphene based materials and a brief background about its catalytic updates. Authors can use following references: ChemCatChem, 2015, 7, 1678-1683; J. Coll. Inter. Sci. 2016, 478, 280-287; Int. J. Biol. Macromol. 101, 696–702.
Response: Thanks for the reviewer’s suggestion and we have added some comments to graphene-based heterojunction photocatalysts for Cr(VI) reduction and RhB degradation in the revised manuscript (Line106-117). We have referred all the paper either in introduction or supporting information.
3. I think the SEM image shows that WO3 is bigger than rGO, while the authors in their schemes have mentioned that rGO. So I think authors should reconsider this matter and give a proof of this claim. I think it is better to measure the particle size of WO3 through TEM image. And please bring a understandable image of SEM.
Response: We are grateful to the reviewer’s thoughtful comments and we are sorry for bothering you with the unclear figure and illustration. We have further marked the SIS, rGO and WO3 nanosheet to illustrate the SEM and HRTEM image (Figure 2) and we have added two SEM images of WGS-5% in the supporting information (Figure S1 E-F, ESI†). The smooth and wrinkle nanosheet in Figure 2A are assigned to the SIS and rGO, respectively. Both of them exhibit the layered structure with various sizes. Actually, it is not easy to identify the WO3 in SEM image, whose size is smaller than the rGO and SIS nanosheet. As shown in Figure 2E, the particle size of WO3 is 40 nm (Line157-167).
4. English also needs to be polished.
Response:We are sorry for bothering you with this kind of mistakes that we could have avoided. We have reviewed and re-checked throughout the whole manuscript, and spelling mistakes, syntactic, and grammatical errors have been eliminated wherever possible in our revised version to address this issue. Thanks.

Reviewer 2 Report
Review of Manuscript ID: catalysts-442826
The authors presented the results of preparation of photocatalyst and its application for Cr(VI) and dye purification. Generally, the paper is worth to be published in Catalysts after major revision. Such approach of dyes removal and Cr(VI) reduction (using photocatalyst) is widely described in literature. It is a pity that the authors did not refer to work related to this topic. In my opinion, the authors should indicate what catalysts have been proposed so far and what results of the decolarisation/reduction have been obtained and indicate the differences between their proposition and the other solution described in literature.
In the paper catalysts have been characterized by XRD, XPS, SEM, TEM, HRTEM and UV-VIS methods. Unfortunately, their application in dye removal and chromium removal has been described too briefly. Some experiments should be added to improve the quality of the paper, such as the reusability of the catalyst. The stability of the catalyst and their recycle are very important parameters from the practical point of view. It will be helpful in results interpretation to check the adsorption ability of the catalysts for dyes herein due to its significant effect on the photocatalytic activity of the catalyst.
In photocatalyst process, one of the main parameter is the ratio concentration of compounds to the amount of photocatalyst. Authors should explain why the initial concentrations of RhB and Cr(VI) was different and why this was used in experimental.
Authors should add more information about analytical method for Cr determination. It is important to know if this method allow to Cr speciation.
Efb cannot be determined directly by the intercept on the V axis. According to the equation 3 calculation of this parameter also requires knowledge of the slope of 1/Csc against V. Please give full calculation methodology.
There are some mistakes in the manuscript: i. description of the y-axis in figure 5b did not correspond to the title of the graph; ii. in the text the explanation should be found when using the abbreviation for the first time (i.e. RhB, TA, BQ, IPA, AO) - the explanation is only at the end of the text, in the experimental part; iii. minor Errors in the literature: 30, 44, 45.
Author Response
Reviewer #2:
1. The authors presented the results of preparation of photocatalyst and its application for Cr(VI) and dye purification. Generally, the paper is worth to be published in Catalysts after major revision. Such approach of dyes removal and Cr(VI) reduction (using photocatalyst) is widely described in literature. It is a pity that the authors did not refer to work related to this topic. In my opinion, the authors should indicate what catalysts have been proposed so far and what results of the decolarisation/reduction have been obtained and indicate the differences between their proposition and the other solution described in literature.
Response:We thank the reviewer for this recommendation. According to the literature, we have further explained the enhanced photocatalytic activity of the as-prepared photocatalyst (Line 239-246 and Line 257-265). Moreover, we have compared the photocatalytic activity of WGS nanohybirds with previous report of other graphene-based Z-scheme heterojunction, and the results were listed in Table S2 (ESI†).
2. In the paper catalysts have been characterized by XRD, XPS, SEM, TEM, HRTEM and UV-VIS methods. Unfortunately, their application in dye removal and chromium removal has been described too briefly. Some experiments should be added to improve the quality of the paper, such as the reusability of the catalyst. The stability of the catalyst and their recycle are very important parameters from the practical point of view. It will be helpful in results interpretation to check the adsorption ability of the catalysts for dyes herein due to its significant effect on the photocatalytic activity of the catalyst.
Response: We are grateful to the reviewer’s thoughtful comments and we have incorporated the recycle test and the XRD analysis of the photocatalyst (Figure S3-S4, ESI†). The diffraction peak intensity of XRD pattern of the sample after 4 times used remain almost unchanged compared to that of the fresh one, indicating the high stability of WGS nanohybrids.
3. In photocatalyst process, one of the main parameter is the ratio concentration of compounds to the amount of photocatalyst. Authors should explain why the initial concentrations of RhB and Cr(VI) was different and why this was used in experimental.
Response: We are grateful to the reviewer’s thoughtful comments. It is suggested that photocatalytic activity is depends on the ability of adsorption capacity of photocatalyst. The point of zero charge of the as-synthesized photocatalyst is below pH 4 and the solution of pH was adjusted to 4, hence the surface of the photocatalyst shows negative charge. Both RhB and Cr(VI) can adsorbed on the surface of photocatalyst due to the electrostatic effect. However, the incorporation of the rGO has significant enhanced adsorption of RhB compared to Cr(VI). Hence the initial concentration of RhB and Cr(VI) were given 30 and 20 mg·L-1 , respectively.
4. Authors should add more information about analytical method for Cr determination. It is important to know if this method allow to Cr speciation.
Response:We are grateful to the reviewer’s thoughtful comments. The concentration of Cr(VI) was determined by the diphenylcarbazide colorimetric method at the maximum absorption wavelength of 540 nm using a UV-vis spectrophotometer(Line395-398).
5. Efb cannot be determined directly by the intercept on the V axis. According to the equation 3 calculation of this parameter also requires knowledge of the slope of 1/Csc against V. Please give full calculation methodology.
Response: We are grateful to the reviewer’s thoughtful comments and we have calculated Efb according the equation of Efb = V - kBT/e (Line226-230 and Table S3).
6. There are some mistakes in the manuscript: i. description of the y-axis in figure 5b did not correspond to the title of the graph; ii. in the text the explanation should be found when using the abbreviation for the first time (i.e. RhB, TA, BQ, IPA, AO) - the explanation is only at the end of the text, in the experimental part; iii. minor Errors in the literature: 30, 44, 45.
Response: We are grateful to the reviewer’s thoughtful comments and we are sorry to make the mistakes in our text. We have revised all the mistakes and we hope the revised manuscript meets the publication standards of the Journal. Thanks.

Reviewer 3 Report
Catalysts
Manuscript ID: catalysts-442826
Title: Z-schemed WO3/rGO/SnIn4S8 sandwich nanohybrids for efficient visible light photocatalytic water purification.
Reviewer comments:
The authors have produced an interesting and complete study concerning the preparation of new photocatalytic nanohybrids. The work has been undertaken diligently and the interpretation of the data supports the conclusions. However, some points should be modified and integrated. I would suggest publishing it on Catalysts after minor revisions.
(1) The authors should uniform the temperature values; “n K” should be “n °C”.
(2) line 149-151: The authors said “In our case, the hierarchical structure of WO3 was further heated at 873 K to obtain a regular nanosheet with a thickness of 14 – 57 nm, which is an ideal interlayer for sandwich.”. The authors should explain the meaning of the expression “ideal interlayer”….ideal for the structure and/or for the thickness obtained.
(3) line 155: The sentence is incomplete.
(4) Figure 2: It is very difficult to identify the layers indicated in pictures C and D…The authors should revise Figure 2, improving the quality of each picture.
(5) Introduction and section 2.2: The authors should explain the choice to study the degradation of RhB and the reduction of Cr(VI)….For this kind of studies, it is usual the use of more common contaminants such as drugs. The authors should add a short explanation about the importance of RhB and Cr(VI) detection.
(6) Sections 2.2 and 2.3: The authors should provide some results or data about recycle tests of the nanohybrid material proposed in this work.
(7) The authors should provide some prospective applications of the new material studied; for example, for which kind of device, in which operative conditions, etc.
Author Response
Reviewer #3:
The authors have produced an interesting and complete study concerning the preparation of new photocatalytic nanohybrids. The work has been undertaken diligently and the interpretation of the data supports the conclusions. However, some points should be modified and integrated. I would suggest publishing it on Catalysts after minor revisions.
1. The authors should uniform the temperature values; “n K” should be “n °C”.
Response: We are grateful to the reviewer’s thoughtful suggestions. The entire temperature unit in the form of “K” has been replaced to meet the publication standards of the Journal.
2. line 149-151: The authors said “In our case, the hierarchical structure of WO3 was further heated at 873 K to obtain a regular nanosheet with a thickness of 14 – 57 nm, which is an ideal interlayer for sandwich.”. The authors should explain the meaning of the expression “ideal interlayer”….ideal for the structure and/or for the thickness obtained.
Response: We are grateful to the reviewer’s thoughtful comments and we are sorry to use the inappropriate expression. We have revised the sentence and we further explained the role of WO3 in the hierarchical structure (Line163-167).
3. line 155: The sentence is incomplete.
Response: We are sorry to make the grammar mistakes and we have revised the sentence (Line168-169).
4. Figure 2: It is very difficult to identify the layers indicated in pictures C and D…The authors should revise Figure 2, improving the quality of each picture.
Response: We are grateful to the reviewer’s thoughtful comments and we are sorry not to provide the figure with enough resolution. Figure 2 has been replaced with enough resolution to meet the publication standards of the Journal
5. Introduction and section 2.2: The authors should explain the choice to study the degradation of RhB and the reduction of Cr(VI)….For this kind of studies, it is usual the use of more common contaminants such as drugs. The authors should add a short explanation about the importance of RhB and Cr(VI) detection.
Response: Thanks for the reviewer’s suggestion and we have added some comments to graphene-based heterojunction photocatalysts for Cr(VI) reduction and RhB degradation in the revised manuscript (Line106-117).
6. Sections 2.2 and 2.3: The authors should provide some results or data about recycle tests of the nanohybrid material proposed in this work.
Response: We are grateful to the reviewer’s thoughtful comments and we have incorporated the recycle test and the XRD analysis of the photocatalyst(Figure S3-S4, ESI†). The diffraction peak intensity of XRD pattern of the sample after 4 times tests remained almost unchanged compared with that of the fresh one, indicating that the high stability of WGS-2.5%.
7. The authors should provide some prospective applications of the new material studied; for example, for which kind of device, in which operative conditions, etc.
Response: We are grateful to the reviewer’s thoughtful comments and we have incorporated the prospective applications of the Z-scheme heterojunction photocatalysts, e.g. water purification, energy conversion, and organic synthesis. Thanks.

Round 2
Reviewer 1 Report
The authors have carefully followed the comments and it is ready for publication in the journal.
Reviewer 2 Report
After correction the masnuscript is worth to be published in Catalysts.